# COVID-19 Vaccines Safety Tracking (CoVaST): Protocol of a Multi-Center Prospective Cohort Study for Active Surveillance of COVID-19 Vaccines’ Side Effects

**DOI:** 10.3390/ijerph18157859

**Published:** 2021-07-25

**Authors:** Abanoub Riad, Holger Schünemann, Sameh Attia, Tina Poklepović Peričić, Marija Franka Žuljević, Mikk Jürisson, Ruth Kalda, Katrin Lang, Sudhakar Morankar, Elias Ali Yesuf, Mohamed Mekhemar, Anthony Danso-Appiah, Ahmad Sofi-Mahmudi, Giordano Pérez-Gaxiola, Arkadiusz Dziedzic, João Apóstolo, Daniela Cardoso, Janja Marc, Mayte Moreno-Casbas, Charles Shey Wiysonge, Amir Qaseem, Anna Gryschek, Ivana Tadić, Salman Hussain, Mohammed Ahmed Khan, Jitka Klugarova, Andrea Pokorna, Michal Koščík, Miloslav Klugar

**Affiliations:** 1Czech National Centre for Evidence-Based Healthcare and Knowledge Translation (Cochrane Czech Republic, Czech EBHC: JBI Centre of Excellence, Masaryk University GRADE Centre), Institute of Biostatistics and Analyses, Faculty of Medicine, Masaryk University, Kamenice 5, 62500 Brno, Czech Republic; abanoub.riad@med.muni.cz (A.R.); mohammad.hussain@med.muni.cz (S.H.); klugarova@med.muni.cz (J.K.); apokorna@med.muni.cz (A.P.); 2Department of Public Health, Faculty of Medicine, Masaryk University, Kamenice 5, 62500 Brno, Czech Republic; koscik@med.muni.cz; 3Department of Health Research Methods, Evidence, and Impact, McMaster University Health Sciences Centre, Room 2C16, 1280 Main Street West, Hamilton, ON L8N 4K1, Canada; schuneh@mcmaster.ca; 4Department of Oral and Maxillofacial Surgery, Justus-Liebig-University, Klinikstrasse 33, 35392 Giessen, Germany; sameh.attia@dentist.med.uni-giessen.de; 5Department of Research in Biomedicine and Health, University of Split School of Medicine, Šoltanska 2, 21000 Split, Croatia; tina.poklepovic@mefst.hr; 6Department of Medical Humanities, University of Split School of Medicine, Šoltanska 2, 21000 Split, Croatia; marija.franka.zuljevic@mefst.hr; 7Department of Family Medicine and Public Health, University of Tartu, Ravila 19, 50411 Tartu, Estonia; mikk.jurisson@ut.ee (M.J.); ruth.kalda@ut.ee (R.K.); katrin.lang@ut.ee (K.L.); 8Faculty of Public Health, Institute of Health, Jimma University, Aba Jifar 1 Street, 378 Jimma, Ethiopia; morankarsn@yahoo.com (S.M.); elias.yesuf@gmail.com (E.A.Y.); 9Clinic for Conservative Dentistry and Periodontology, School of Dental Medicine, Kiel University, Arnold-Heller-Str. 3, Haus B, 24105 Kiel, Germany; mekhemar@konspar.uni-kiel.de; 10Department of Epidemiology and Disease Control, School of Public Health, University of Ghana, Accra LG 25, Ghana; adanso-appiah@ug.edu.gh; 11Cochrane Iran Associate Centre, National Institute for Medical Research Development, Tehran 16846, Iran; sofimahmudi@research.ac.ir; 12Department of Evidence Based Medicine, Sinaloa Pediatric Hospital, Cochrane Mexico, Calle Constitución 530, Jorge Almada, 80200 Culiacán, Mexico; giordano@cochrane.mx; 13Department of Conservative Dentistry with Endodontics, Medical University of Silesia, 40-055 Katowice, Poland; adziedzic@sum.edu.pl; 14Health Sciences Research Unit: Nursing, Portugal Centre for Evidence-Based Practice: JBI Centre of Excellence, Nursing School of Coimbra, 3004-011 Coimbra, Portugal; apostolo@esenfc.pt (J.A.); dcardoso@esenfc.pt (D.C.); 15Faculty of Pharmacy, University of Ljubljana, Aškerčeva Cesta 7, 1000 Ljubljana, Slovenia; janja.marc@ffa.uni-lj.si; 16Institute of Clinical Chemistry and Biochemistry, University Medical Centre Ljubljana, Zaloška cesta 2, 1000 Ljubljana, Slovenia; 17Nursing and Healthcare Research Unit (Investén-isciii), Instituto de Salud Carlos III, 28029 Madrid, Spain; mmoreno@isciii.es; 18Cochrane South Africa, South African Medical Research Council, Cape Town 7501, South Africa; charles.wiysonge@mrc.ac.za; 19American College of Physicians, 190 N Independence Mall W, Philadelphia, PA 19106, USA; aqaseem@acponline.org; 20Department of Nursing in Collective Health, School of Nursing, University of São Paulo, São Paulo 419, Brazil; gryschek@usp.br; 21Department of Social Pharmacy and Pharmaceutical Legislation, Faculty of Pharmacy, University of Belgrade, 11221 Belgrade, Serbia; ivana.tadic@pharmacy.bg.ac.rs; 22School of Pharmaceutical Education and Research, Jamia Hamdard, Hamdard Nagar, New Delhi 110062, India; drm.ahmedkhan@jamiahamdard.ac.in; 23Institute of Health Information and Statistics of the Czech Republic, Palackého náměstí 375/4, 128 01 Praha 2, Czech Republic; 24Department of Nursing and Midwifery, Faculty of Medicine, Masaryk University, Kamenice 5, 625 00 Brno, Czech Republic; 25Czech Clinical Research Infrastructure Network, Department of Pharmacology, Faculty of Medicine, Masaryk University, Kamenice 5, 625 00 Brno, Czech Republic

**Keywords:** cohort studies, cross-sectional studies, COVID-19, drug-related side effects and adverse reactions, health personnel, mass vaccination, prevalence

## Abstract

Background: Coronavirus disease (COVID-19) vaccine-related side effects have a determinant role in the public decision regarding vaccination. Therefore, this study has been designed to actively monitor the safety and effectiveness of COVID-19 vaccines globally. Methods: A multi-country, three-phase study including a cross-sectional survey to test for the short-term side effects of COVID-19 vaccines among target population groups. In the second phase, we will monitor the booster doses’ side effects, while in the third phase, the long-term safety and effectiveness will be investigated. A validated, self-administered questionnaire will be used to collect data from the target population; Results: The study protocol has been registered at ClinicalTrials.gov, with the identifier NCT04834869. Conclusions: CoVaST is the first independent study aiming to monitor the side effects of COVID-19 vaccines following booster doses, and the long-term safety and effectiveness of said vaccines.

## 1. Introduction

Coronavirus disease (COVID-19) mass vaccination has been a chief priority for health systems globally, which needs to be accelerated in order to control the acute phase of the pandemic [1,2]. Nevertheless, vaccine hesitancy (VH)—which refers to the “delay in acceptance or refusal of vaccines despite the availability of vaccine services”—remains a serious challenge for vaccination strategies worldwide [3,4,5]. In 2019, the World Health Organization (WHO) declared VH as one of the top 10 global health threats, which is nourished by misinformation regarding vaccines’ effectiveness and safety [6].

Aversion to vaccines’ potential side effects is the most frequent cause of VH among various population groups [7,8]. Therefore, a recent systematic review revealed that raising public awareness of vaccines’ effectiveness, and honesty regarding their side effects, are vital strategies to improve vaccine uptake [9].

According to the Strategic Advisory Group of Experts on Immunization of the WHO (SAGE-WHO), distrust in the pharmaceutical industry is a contextual driver of VH, because vaccine manufacturers can be perceived as preferring their financial benefit over public health interest [10]. In both high-income and low-income settings, distrust of the pharmaceutical industry has been consistently and significantly higher among hesitant groups, and this is aggravated by a lack of transparency regarding public health plans [11,12,13].

Public health systems currently experience a novel and a unique challenge, due to the variety of vaccine manufacturers, and the high levels of public awareness about those manufacturers and their marketing strategies [8]. This unprecedented situation is predicted to create what we can refer to as “vaccine selectivity”, where individuals can prefer a certain type or brand of vaccine over others; this situation will increase the pressure on our weakened health systems and economies, and of course it can increase the VH levels as well [14].

The temporary suspension of the Oxford–AstraZeneca vaccine (AZD1222) (AstraZeneca plc, Cambridge, UK) and the Janssen vaccine (Ad26.COV2.S) (Johnson & Johnson (J&J), New Brunswick, NJ, US) due to reports of extremely rare side effects triggered public debates that might have adversely affected vaccination acceptance levels [15,16]. However, although the European and American cdrug regulators declared that the benefits of using these vaccines still outweigh their risks, very little is known about vaccine hesitancy and, probably, selectivity after these incidences [15,17].

Given the projected seasonality of COVID-19 transmission and the increasing number of its variants, vaccine manufacturers launched trials for booster doses that are predicted to be readily available by the fall of 2021 [18,19,20].

Independent (non-sponsored) studies with rigorous methods can successfully lead the unbiased pharmacovigilance efforts of COVID-19 vaccines globally [21,22,23,24,25,26,27,28]. Thus, in view of their independent nature and transparent design, these studies can play a key role in suppressing VH levels by enhancing public confidence in the vaccines.

### Objectives

This project aims to actively monitor the side effects and effectiveness of COVID-19 vaccines worldwide. The primary objectives of the project include:(a)To estimate the prevalence of both local and systemic side effects following each of the COVID-19 vaccines among healthcare workers (HCWs), teachers and academics (TAs), senior adults ≥65 years old (SAs), and minors ≤18 years old (MIs);(b)To evaluate the potential demographic and medical risk factors for the frequency and intensity of side effects;(c)To evaluate the long-term safety of COVID-19 vaccines.

The secondary objectives include:(a)To evaluate the relative effectiveness and safety of COVID-19 vaccines in relation to one another;(b)To evaluate the impact of palliative medications used by the vaccinated individuals for short-term side effect resolution.

## 2. Materials and Methods

### 2.1. Design

This project is composed of three main phases: (a) a cross-sectional survey for the short-term side effects of COVID-19 vaccines; (b) a prospective cohort study for the safety of COVID-19 vaccines following booster doses; and (c) a prospective cohort study for the long-term safety and effectiveness of COVID-19 vaccines.

#### 2.1.1. Phase A

A validated, self-administered questionnaire will be developed and delivered online to the target population groups (HCWs, TAs, SAs, and MIs). In certain circumstances, telephone interviews and paper questionnaires will be used instead of the online questionnaire in order to adapt to the local setting. The questionnaire will inquire about the short-term side effects following either the first dose, the second dose, or both doses of the COVID-19 vaccine. The side effects will be classified as local or systemic, and their onset, duration, and intensity will be self-assessed and self-reported by the participating subjects. This phase is planned to take place until 31 December 2021.

#### 2.1.2. Phase B

A validated, self-administered questionnaire will be developed and delivered online to the volunteers who participated in Phase A and expressed their interest in reporting on their long-term outcomes. The short-term side effects following booster doses will be investigated in this phase. This phase is tentatively planned to take place from October 2021 until April 2022.

#### 2.1.3. Phase C

A validated, self-administered questionnaire will be developed and delivered online to the volunteers who participated in Phase A and expressed their interest in self-reporting their long-term outcomes. The vaccines’ effectiveness and safety will be monitored, and this phase will last for five consecutive years, starting from January 2022.

### 2.2. Population

In Phase A, a pragmatic approach will be used, tracking each target population group according to individual governments’ distributional plans, which in most countries went from HCWs, to SAs, to TAs, to MIs. The sample of Phases B and C will be pre-identified based on the outcomes of Phase A.

If more than 368 of the Phase A participants show their interest in joining Phase B, no additional recruitment will be required. If less than 368 of the Phase A participants show their interest in participating in Phase B, additional recruitment will be carried out, targeting HCWs who will receive booster doses. In case of the emergence of special side effects after booster doses, additional recruitment of a sample of HCWs will be required.

#### 2.2.1. Inclusion Criteria

HCWs, TAs, SAs, and MIs who received a COVID-19 vaccine in the post-authorization phase;The recently vaccinated individuals who received their vaccine dose within the previous 30 days will be prioritized to be invited for the study, even though the study will not be limited to the recently vaccinated individuals;Participating subjects should be at least 18 years old in order to give their informed consent independently, or in case of the minors (below 18 years old), their caregivers will be asked to give their informed consent.

#### 2.2.2. Exclusion Criteria

HCWs, TAs, SAs, and MIs who received the COVID-19 vaccines as part of phase III clinical trials.

#### 2.2.3. Sample Size

The pragmatic sample size for each target group in each country will be calculated using Epi Info ^TM^ version 7.2.4 (CDC, Atlanta, GA, USA. 2020). The formula of population survey studies will be used to achieve a 5% margin of error and a 95–99% confidence level [29]. The expected frequency (outcome probability) is assumed to be 60%, as the prevalence of side effects following COVID-19 vaccines ranged between 62% and 93% in our previous studies [21,22] (Figure 1).

### 2.3. Instrument

The questionnaire will be based on the growing evidence of COVID-19 vaccines’ side effects, and adverse reactions and will be updated and validated accordingly. The questionnaire consists of four categories: (a) demographic data (age, gender, height, weight, profession, and geographic region); (b) medical anamneses (chronic illnesses, medications, smoking, and alcohol consumption); (c) COVID-19-related anamneses (type of vaccine, number of vaccine doses, dates of vaccine doses, previous infection, and diagnosis date); and (d) vaccine side effects (local side effects, systemic side effects, onset, and duration) Appendix A.

The multi-linguistic versions of the instrument will be produced through a pragmatic workflow for translation and cultural adaptation [31]. The current instrument is designed and validated for the HCWs group. The instrument will be validated for the other two populations of interest (OAs and TAs) by a validation process using a panel of experts, with four experts from the targeted population and four experts with a background in public health, epidemiology, infectious disease, and vaccination.

Two native speakers of the target language with a high level of English proficiency will translate the instrument independently. An expert panel composed of three members (the two forward translators, and a third native speaker with a biomedical background and an advanced grasp of the English language) will review the two translated versions, and will resolve discrepancies between them, aiming to generate a harmonized final version. The working version will undergo reliability testing through test–retest. In the test–retest, a minimum of 10 volunteers should fill in the questionnaire twice, at least two weeks apart.

### 2.4. Recruitment

Data will be collected in two phases via an online, validated, self-administered questionnaire. Although the data collection strategies may differ across the globe, the target groups are recommended to be approached by governmental, professional, and university networks.

#### 2.4.1. Phase A

A.1. HCWs will be approached by medical and healthcare chambers and/or healthcare professional organizations, and the snowballing technique will be applied;

A.2. Senior adults (≥65) will be approached by the “university of the third age”, by the professional medical association of “young general practitioners”, and by professional organizations for older adults, and the snowballing technique will be applied;

A.3. School teachers will be approached by the networks of educational institutions, while university teachers will be approached via all major universities, and the snowballing technique will be applied;

A.4. Minors (≤18) will be principally approached through their schools, where their parents (or guardians) will be invited to fill in the questionnaire on behalf of their children.

Data collection for the A.1., A.2., A.3., and A.4. population groups will be adjusted according to each participating country’s local setting.

#### 2.4.2. Phase B and C

Volunteers who participate in Phase A and express their interest in self-reporting their long-term side effects will be approached again. The vaccine effectiveness and side effects following booster doses will be investigated in Phase B. Phase C will take place for five consecutive years, starting from 2022.

### 2.5. Timeline

As the local timelines are dependent on the setting of each participating country—including governments’ distribution plans, availability of vaccines, and administrative processes—the proposed timeline is deemed to guide the overall CoVaST progress (Table 1).

### 2.6. Ethics

The study was reviewed and approved by the Ethics Committee of the Faculty of Medicine at Masaryk University on 19 May 2021 (Ref. 26/2021). Ethical clearance will be secured from a designated institutional review board in each participating country before commencement of the study.

Digital informed consent will be obtained from each participant prior to participation. The participants will be allowed to withdraw from the study at any moment without justification, and no data will be saved before the participants submit their answers completely.

### 2.7. Analysis

Descriptive statistics will be performed to check the normality of data distribution, and to present the frequencies and percentages of dependent variables (side effects) and independent variables (demographic data, medical anamneses, and COVID-19-related anamneses). Inferential statistics will be performed to evaluate the potential association of each side effect and the suggested demographic and medical risk factors. All tests will be performed using SPSS 27 (IBM Corp, Armonk, NY, USA), and the significance level cutoff will be set at *p* ≤ 0.05 [32].

## 3. Registration and Dissemination

The study protocol has been registered with the US National Library of Medicine registry (ClinicalTrials.gov, accessed on 9 May 2021), with the identifier NCT04834869 [33]. The ClinicalTrials.gov (accessed on 9 May 2021) record will be regularly updated by the project’s principal investigators, and any deviations from the protocol will be mentioned and justified a priori in the electronic record, and in the manuscript of the final study.

The investigators aim to disseminate the results of the project in peer-reviewed journals on a regular basis. For the results of phase A, the international data for each target group will be published once the data collection is completed. Meanwhile, national data of each participating center will be published as interim results. For the results of phase B, the international data for each target group will be published once the data collection is completed. For the results of phase C, the international data for all target groups will be published on an annual basis once the data collection is completed.

## 4. Discussion

Post-marketing evaluation of vaccines’ safety has typically relied on voluntary reporting of side effects by health care professionals, vaccinated individuals, and caregivers. While there is a surging demand for rigorous pharmacovigilance systems, with active surveillance designs rather than the traditional passive surveillance, a very limited number of high-income countries have managed to develop such systems so far [34].

The United Kingdom (UK) is one of the leading countries in this field, due to its early efforts in developing active surveillance systems for the safety of diphtheria/tetanus/pertussis (DTP) and measles/mumps/rubella (MMR) vaccines since the early 1990s [35]. In terms of COVID-19 vaccines’ safety, the Medicines and Healthcare products Regulatory Agency (MHRA) of the UK has adopted an innovative hybrid system that includes: (a) enhanced passive surveillance through the Yellow Card scheme, where members of the public and healthcare professionals voluntarily report suspected side effects; (b) targeted active monitoring using the Yellow Card scheme; and (c) formal epidemiological studies, such as the OpenSAFELY 17 Collaborative and COVID Symptom Study app [25,36,37].

The results of post-marketing studies may differ to various degrees from the outcomes of phase III trials, where apparently healthy volunteers are usually recruited following strict criteria. Riad et al. found that the overall prevalence of Pfizer–BioNTech ( (Pfizer Inc.: New York City, US) COVID-19 vaccine side effects among recently vaccinated HCWs in the Czech Republic was relatively higher than those reported by the manufacturer [21]. Similarly, the side effects of the Moderna COVID-19 vaccine and the CoronaVac vaccine were more prevalent among HCWs in the US and Turkey, respectively, than the manufacturers’ reports [22,27]. On the other hand, Menni et al. found that the side effects of the Pfizer–BioNTech and Oxford–AstraZeneca COVID-19 vaccines occurred less frequently among a large cohort in the UK than those reported in the phase III trials [25].

The demographic and medical risk factors for side effects’ frequency and intensity are not usually reported by phase III trials, as they are not necessarily outcomes of interest during this stage. Therefore, post-marketing studies are in an ideal position to confirm or refute suggested risk factors, using large datasets of self-reported outcomes. For example, all post-marketing studies of mRNA vaccines found that the frequency of side effects following the second dose is higher than after the first dose [21,23,24,25]. The phase III trials displayed the same pattern; therefore, the post-marketing studies only came to confirm this preliminary finding [38,39]. Female gender was consistently associated with an increased risk of side effects following different types of COVID-19 vaccines; interestingly, the gender-based differences were reported by manufacturers [21,22,23,25,28].

As more COVID-19 vaccines are currently in the pipeline of clinical trials and authorization, readily available instruments for active surveillance will be much needed in order to shorten the time period of post-marketing investigation by academic institutions. Moreover, the prospective booster doses’ safety should be evaluated in relatively shorter periods of time, in order to relieve our weakened healthcare systems. Therefore, the CoVaST project aims to provide an international infrastructure for active surveillance of booster doses’ side effects and the long-term safety and effectiveness of COVID-19 vaccines.

### Strengths and Limitations

To the best of the authors’ knowledge, this is the first multinational study aiming to monitor the safety of various COVID-19 vaccines, especially following booster doses. Another strong point of this study is its unified evaluation instrument, target groups, and methods, which will be used in all participating countries, and should maximize the results’ internal validity. Recruiting HCWs is intended to limit the reporting bias that is naturally predicted in this survey-based study, due to the fact that HCWs retain high levels of health literacy and scientific interest. This study is one of the early-registered studies that are concerned with the long-term safety of COVID-19 vaccines, and their effectiveness.

In general, this study is limited by the heterogeneous time span between vaccination and survey commencement across the participating countries; therefore, subgroup analysis according to the time span will be carried out during data analysis. One more limitation is recall bias, as in various countries, vaccination covered the majority of population who intended to be vaccinated. Due to the recruitment of participants who received the vaccine in the first half of 2021, there is a possibility of recall bias when filling out the questionnaire. Since COVID vaccination is a hot topic worldwide, we assume that participants could remember all experienced side effects well.

## 5. Conclusions

The side effects of COVID-19 vaccines require active surveillance in the post-authorization phase, as the side effects can potentially impact decisions regarding vaccination. CoVaST, as a multi-national study, aims to evaluate the short-term and long-term side effects and effectiveness of various COVID-19 vaccines.

## Figures and Tables

**Figure 1 ijerph-18-07859-f001:**
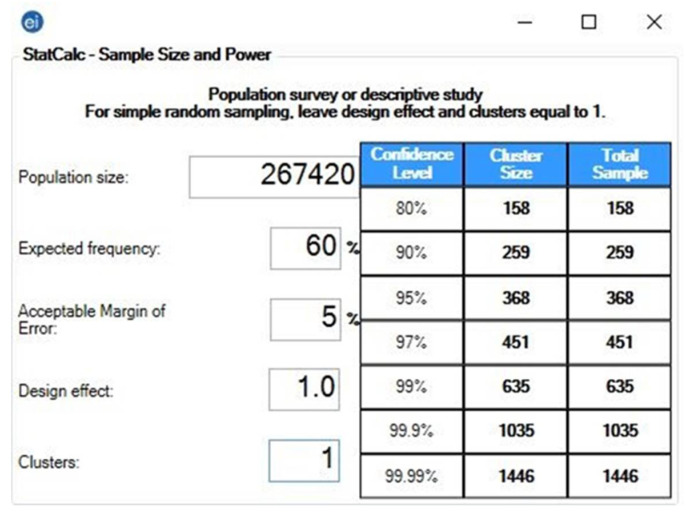
Sample size of healthcare workers (HCWs) in the Czech Republic—Epi-Info ^TM^ version 7.2.4. Population size: Total number of healthcare workers in the Czech Republic in 2017 [30]. Expected frequency: The overall prevalence of side effects following COVID-19 vaccines ranged between 62% and 93%; therefore, 60% was assumed as a threshold. Acceptable margin of error: The permissible level for all CoVaST groups will be 5%. Design effect: One—per the recommendation of the CDC for simple sampling [29]. Clusters: One—per the recommendation of the CDC for simple sampling [29]. The pragmatic sample size is 368–635 (CI 95%–99%).

**Table 1 ijerph-18-07859-t001:** The projected timeline of the COVID-19 vaccines safety tracking (CoVaST) study.

Phase	Stage	Population	Schedule
Phase A	Stage A.1.	HCWs	May–August 2021
	Stage A.2.	SAs	June–December 2021
	Stage A.3.	TAs	June–December 2021
	Stage A.4.	MIs	June–December 2021
Phase B	Stage B.1.	HCWs	October 2021–February 2021
	Stage B.2.	SAs	November 2021–April 2022
	Stage B.3.	TAs	November 2021–April 2022
	Stage B.4.	MIs	November 2021–April 2022
Stage C	Stage C.1.	HCWs, SAs, TAs, MIs	January–December 2022
	Stage C.2.	HCWs, SAs, TAs, MIs	January–December 2023
	Stage C.3.	HCWs, SAs, TAs, MIs	January–December 2024
	Stage C.4.	HCWs, SAs, TAs, MIs	January–December 2025
	Stage C.5.	HCWs, SAs, TAs, MIs	January–December 2026

HCWs = Healthcare Workers; SAs = Senior Adults; TAs = Teachers and Academics; MIs = Minors.

## Data Availability

The data that support the findings of this study are available from the corresponding author upon reasonable request.

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
