# Peer review of "COVID-19 Vaccines Safety Tracking (CoVaST): Protocol of a Multi-Center Prospective Cohort Study for Active Surveillance of COVID-19 Vaccines’ Side Effects"

_ijerph, 2021, doi:10.3390/ijerph18157859_

Round 1

Reviewer 1 Report

The study “COVID-19 Vaccines Safety Tracking (CoVaST): Protocol of a 2 Multi-center Prospective Cohort Study for Active Surveillance 3 of COVID-19 Vaccines Side Effects" to actively monitor safety and effectiveness of COVID-19 vaccines globally. Post-marking safety assessment is essential in the rapid mass immunization campaigns implemented with the SARS-COV-2 vaccine.  However, some issues need to be addressed.

Major comments

  1. This study involves three primary objectives, so it is necessary to calculate the sample size based on the three primary objectives and select the maximum of them as the sample size for this study to ensure that the three primary research objectives can be met.
  2. What does the questionnaire in Phase B and Phase C contain? What are the main medium- and long-term adverse reactions that this study intends to observe?
  3. The vaccines effectiveness and safety will be monitored in phase C. How will the vaccines effectiveness be assessed at this stage?
  4. In phase A, it is better only included the recently vaccinated individuals who received their vaccine dose within the previous 30 days or 14 days to avoid underestimation of adverse reactions due to recall bias.
  5. The inclusion criteria “Participating subjects should be at least 18-year-old” is not appropriate since minors ≤ 18 years-old will be included in this study.

Author Response

Dear Reviewer

We are delighted to have the opportunity to revise and resubmit our manuscript titled “COVID-19 Vaccines Safety Tracking (CoVaST): Protocol of a Multi-center Prospective Cohort Study for Active Surveillance of COVID-19 Vaccines Side Effects” (Manuscript ID ijerph-1257063). 

We have considered all remarks provided by all of the reviewers. Please find appended a revised version of the manuscript (with track-changes highlighted) and a point-by-point rebuttal to all comments raised as detailed below. We hope our responses are satisfactory in addressing the criticisms and suggestions.

We hope the revised manuscript will be in acceptable format. Thank you for your kind contribution.

  1. This study involves three primary objectives, so it is necessary to calculate the sample size based on the three primary objectives and select the maximum of them as the sample size for this study to ensure that the three primary research objectives can be met.

Answer

We would like to thank the reviewer for this recommendation. We have modified the paragraph regarding the sample size of each phase line 168 - 173. The prgamatic (minmum) sample size of 368 participants should be achieved in each phase and in each target group. In case we do not get potential 368 participants for the phase B and C, we will do additional recruitment.

2. What does the questionnaire in Phase B and Phase C contain? What are the main medium- and long-term adverse reactions that this study intends to observe?

Answer

We would like to thank the reviewer for raising this interesting point and giving us the opportunity to clarify it. At the moment we do not have a solid idea for the questions that should be included in the long-term survellience, because our knowledge about the COVID-19 vaccines safety is still growing. In fact, we will develop the questionnaires of C according to the results we will get from phase A. Moreover, we will be inquiring about the long-term safety that will be used by antivaxxer campaigns to avert public to vaccination. For example, we may ask questions about thrombotic events, fretility, etc. The booster doses questionnaire (phase B) will be more likely similar to the phase A questionnaire; however, we need to wait and see the results of phase A that will inform our next phases.

3. The vaccines effectiveness and safety will be monitored in phase C. How will the vaccines effectiveness be assessed at this stage?

Answer

We would like to thank the reviewer for raising this interesting point. We will ask a simple Yes/No question whether they were infected by SARS-CoV-2 after reciving the vaccine. If (Yes), we will then ask about the timing, means of confirmation of infection and its symptoms.

4. In phase A, it is better only included the recently vaccinated individuals who received their vaccine dose within the previous 30 days or 14 days to avoid underestimation of adverse reactions due to recall bias.

Answer

We would like to thank the reviewer for this recommendation. In our recent studies on COVID-19 vaccine side effects, we found that recall bias has no or very minimal effect on the accuracy of reported information by the participants. We checked the test re-test reliabilty with intervals of 2-3 weeks, and the participants filled in the questionnaire (0-59) days after vaccination. The inter-rater agreement was really high. Please refer to our previous studies: https://pubmed.ncbi.nlm.nih.gov/33916020/

Moreover, some side effects may last for more than 4 weeks, therefore, it would be safer to avoid this time limit condition.

5. The inclusion criteria “Participating subjects should be at least 18-year-old” is not appropriate since minors ≤ 18 years-old will be included in this study.

Answer

We would like to thank the reviewer for this recommendation. We have amended our manuscript accordingly. Line 180 - 182

Sincerely,

Reviewer 2 Report

The authors suggest a cohort study for surveillance of side effects of COVID-19 vaccination. Both short-term and long-term side effects will be targeted. I am sure that different groups are currently tracing side effects of COVID-19 vaccination, but I do agree with the authors that a unified evaluation method can facilitate comparison of the results obtained by different groups in different countries. This is the major strength of the manuscript. This manuscript also provides a tool for countries that do not have similar surveillance program. I just would like to inquire how frequent the subjects will be asked for reporting the long-term side effects? In addition, do the authors have any plan to disseminate the findings (frequency and medium for dissemination)? I believe effective dissemination of the findings would help to reduce vaccine hesitancy. Otherwise, I identify no major flaws of this manuscript.

Author Response

Dear Reviewer

We are delighted to have the opportunity to revise and resubmit our manuscript titled “COVID-19 Vaccines Safety Tracking (CoVaST): Protocol of a Multi-center Prospective Cohort Study for Active Surveillance of COVID-19 Vaccines Side Effects” (Manuscript ID ijerph-1257063).

We have considered all remarks provided by all of the reviewers. Please find appended a revised version of the manuscript (with track-changes highlighted) and a point-by-point rebuttal to all comments raised as detailed below. We hope our responses are satisfactory in addressing the criticisms and suggestions.

We hope the revised manuscript will be in acceptable format. Thank you for your kind contribution.

  1. I just would like to inquire how frequent the subjects will be asked for reporting the long-term side effects? In addition, do the authors have any plan to disseminate the findings (frequency and medium for dissemination)? I believe effective dissemination of the findings would help to reduce vaccine hesitancy. Otherwise, I identify no major flaws of this manuscript.

Answer

We would like to thank the reviewer for raising this interesting point and giving us the opportunity to clarify it. We have added a paragraph about our dissemination plan line 274 - 281.

     The investigators aim to disseminate the results of the project in peer-reviewed journals on regular basis. For the results of phase A, the international data of each target group will be published once the data collection is completed. Meanwhile, national data of each participating centre will be published as interim results. For the results of phase B, the international data of each target group will be published once the data collection is completed. For the results of phase C, the international data of all target groups will be published on an annual basis once the data collection is completed.

Sincerely,

Round 2

Reviewer 1 Report

The authors have completed most of the revisions according to the reviewers' comments. However, one issue remains to be clarified. 

The objective of phase C is to evaluate the long-term safety of COVID-19 vaccines. Currently, the sample size is based on the assumption the overall prevalence of side effects is 60%. However, long-term safety often focuses on rare or delayed adverse reactions, and the prevalence is relatively low. We suggest the sample size of phase C should be calculated according to the prevalence of rare or delayed adverse reactions. The sample size of 368 is not enough for phase C.

Author Response

Dear Reviewer

We would like to thank you for raising this interesting point and giving us the opportunity to clarify it.

  • To the best of our knowledge, there is no vaccine in general (not just COVID-19 vaccines) that was found to be really associated with long-term side effects. According to the CDC's standards, the vaccines which impose no serious side effects within the first six weeks are deemed to be safe in the long term because they will not impose any side effects after that. These standards are based on years of experience with surveillance of various vaccines in the US.
    https://www.cdc.gov/coronavirus/2019-ncov/vaccines/safety/safety-of-vaccines.html
  • Therefore, we are unable to predict that the long-term side effects of COVID-19 vaccines will be very rare due to what we have learnt so far about these vaccines safety profiles from phase A.
  • We do agree with you that in case that there will be rare long-term side effects (prevalence < 10%) we will need to decrease the margin error from 5 to 2 or even less which will increase the sample size from 386 to few thousand. We are prepared for this scenario through the size of our global sample which is predicted to be composed of several thousand participants.
  • The last point we want to bring here is "Why do we need long-term surveillance while we are pretty confident about the long-term safety of COVID-19 vaccines?"
    Public confidence in vaccines safety is an essential element to fight vaccine hesitancy and enhance vaccination strategies; therefore, active and independent surveillance of the long-term safety of the vaccines serve as an important asset in this context. Moreover, we are aware that booster doses and probably annual (seasonal) doses of COVID-19 vaccines may be needed in the future. We will develop the questionnaires of C according to the results we will get from phase A, and we will be inquiring about the long-term safety that will be used by anti-vaxxer campaigns to avert the public from vaccination. For example, we may ask questions about thrombotic events, fertility, etc.

Sincerely,